**Data Availability Statement:** Data are available in ICPSR: (https://www.openicpsr.org/openicpsr/

# Community partners' responses to items assessing stakeholder engagement: Cognitive response testing in measure development

**Vetta L. Sanders Thompson**[1☯*], **Nora Leahy**[2☯], **Nicole Ackermann**[2☯], **Deborah J. Bowen**[3‡], **Melody S. Goodman**[4‡]

**1** Washington University in St. Louis, Brown School, St. Louis, MI, United States of America, **2** Washington University in St. Louis, School of Medicine, St. Louis, MI, United States of America, **3** University of Washington, Department of Bioethics and Humanities, Seattle, WA, United States of America, **4** New York University, School of Global Public Health, New York, NY, United States of America

☯ These authors contributed equally to this work.
‡ These authors also contributed equally to this work.
* vthompson22@wustl.edu

## Abstract

### Background

Despite recognition of the importance of stakeholder input into research, there is a lack of validated measures to assess how well constituencies are engaged and their input integrated into research design. Measurement theory suggests that a community engagement measure should use clear and simple language and capture important components of underlying constructs, resulting in a valid measure that is accessible to a broad audience.

### Objective

The primary objective of this study was to evaluate how community members understood and responded to a measure of community engagement developed to be reliable, valid, easily administered, and broadly usable.

### Method

Cognitive response interviews were completed, during which participants described their reactions to items and how they processed them. Participants were asked to interpret item meaning, paraphrase items, and identify difficult or problematic terms and phrases, as well as provide any concerns with response options while responding to 16 of 32 survey items.

### Results

The results of the cognitive response interviews of participants (N = 16) suggest concerns about plain language and literacy, clarity of question focus, and the lack of context clues to facilitate processing in response to items querying research experience. Minimal concerns were related to response options. Participants suggested changes in words and terms, as well as item structure.

workspace?goToPath=/openicpsr/
126361&goToLevel=project).

**Funding:** MG, VLST - ME-1511-33027 This
research was funded by the Patient Centered
Outcome Research Institute (PCORI) https://www.
pcori.org/ The funder had no role in the study
design, data collection, analysis, interpretation, or
drafting of this article.

**Competing interests:** The authors declare that they
have no competing interests.

## Conclusion

Qualitative research can improve the validity and accessibility of measures that assess
stakeholder experience of community-engaged research. The findings suggest wording and
sentence structure changes that improve ability to assess implementation of community
engagement and its impact on research outcomes.

## Introduction

Policy makers, funders, patients, and providers are increasingly interested in community/
patient-engaged research, which is broadly understood as stakeholder-engaged research. Inter-
est in this work is based on the belief that this strategy contributes to more acceptable research
designs and methods and culturally sensitive and ethical proposals that reduce participant bur-
den and enhance recruitment and retention of participants [1–3]. In addition, stakeholder-
engaged research may facilitate implementation research. These studies require planning for
sustainability after funding, trust among collaborating stakeholders, and ensuring research
capacity of stakeholder partners in order to leverage stakeholder resources and develop recip-
rocal relationships between researchers and stakeholders [1]. To assure that engagement meets
the goals articulated, there is a need to rigorously evaluate the impact of stakeholder engage-
ment on the development, implementation, and outcomes of research studies, in order to
move from lessons learned to evidence-based practices for stakeholder engagement [2].

It is important to understand how the level of engagement in a partnership is developing,
and to what extent the level of engagement is a predictor of outcomes in the study. Rigorous
measurement of engagement requires the development and validation of tools to assess stake-
holder engagement, using items that respondents understand and that measure what they are
intended to measure. A systematic review of measures of stakeholder engagement in research
showed that this area of research is not very strong methodologically [3]. However, we present
in this paper the results of one step in the process of developing a clearly defined, reliable, and
valid measure of stakeholder engagement. Cognitive response interviews were employed to
explore participants' reactions to item wording and response options in addition to their abil-
ity to comprehend items and determine a response. A grounded theory approach guided data
analysis and interpretation.

In addition to item development, item refinement must be considered in measure develop-
ment and validation. Measurement items that work well are defined as those that use clear and
simple language, are not rated as difficult by interviewers or research participants, do not result
in significant missing data, do not yield unexpected frequencies or patterns of association, and
capture an important component of the underlying construct [4]. Studies addressing item and
measurement quality note variations in item interpretation based on participation in the activ-
ity under study and variations in recall based on the frequency of participation in the activity/
activities. Community engagement studies are subject to the types of variations in participa-
tion that may impact item interpretation and recall [5, 6]. Researchers recognize the impor-
tance of the wording of survey questions. One approach to identifying and resolving variations
in item interpretation related to experience, language, and construction is to use cognitive
response interviewing.

Cognitive interview methods have become an important strategy to ensure the quality and
accuracy of survey instruments and are used to identify and analyze sources of response error
in survey questionnaires [7, 8]. Because it helps to explain how people come up with answers

to survey questions, cognitive response interviewing is used to evaluate and improve survey items and design [8]. Willis and Artino note that cognitive response interviewing is a part of a systematic and rigorous survey design process [9]. They note that the process allows researchers to answer the question: "Will my respondents interpret my items in the manner that I intended?" (p. 353) [9]. Cognitive response interviewing is an evidence-based option for examining participant comprehension and interpretation of survey items, as well as respondents' understanding of intended differences in survey items [10, 11].

Two common cognitive interviewing techniques are verbal probing and "think aloud" [8–10]. In verbal probing, participants answer questions about their interpretations of a survey item, paraphrase the survey item, and identify words, phrases, or item components that are problematic. In "think aloud," participants verbalize ideas that come to mind as they answer a question and thereby shed light on reactions, inferences, and beliefs that helped them arrive at their answer. Both techniques are useful in identifying problems with survey item wording and design, but verbal probing is more likely to address issues with "literacy and plain language (i.e., jargon-free and carefully worded language" consistent with the *Federal Plain Language Guidelines* [12]).

This article adds to the literature on the measurement of stakeholder engagement by describing literacy concerns, attitudes about the information needed to judge engagement, as well as response preferences for items used in the public health literature to assess the level, type, and impact of community-engaged interventions and research. In addition, findings may improve the way that researchers communicate with stakeholders about community-engaged research and the assessment of this type of research.

## Methods

### Participant sample

A purposive sample of 16 was recruited to complete one-on-one cognitive response interviews. Eligibility criteria for the cognitive response interviews included being an adult (18 years or older) with experience partnering with researchers on patient- or community-engaged research. Participants were recruited by email from a database of Community Research Fellows Training (CRFT) alumni, who completed the CRFT program in St. Louis, MO, [13] and through referral by CRFT alumni. CRFT was established in 2013 and maintains a voluntary database of graduates of four cohorts (n = 125), 94 (75%) of whom are active alumni and have updated contact information.

### Item selection

In 2011, some of the authors of this paper [14] participated in a review of the community-based participatory research (CBPR) and community engagement literature to determine best practices in evaluating adherence to, effectiveness of, and implementation of CBPR, as well as to identify relevant items. Based on this review, the evaluation team for the Program for the Elimination of Cancer Disparities (PECAD) (including some authors) identified and adapted questions for the survey from published measures on group dynamics, characteristics of effective partnerships, intermediate measures of partnership effectiveness, facilitation of partner involvement, and member satisfaction [15–17]. The original survey (initiated in 2011) contained 60 items and included both closed- and open-ended questions [14].

In 2013, a PECAD evaluation committee was formed and initiated a revision of the original 2011 survey. The goal was to create a measure that was comprehensive and adequately addressed CBPR principles. The developers of the new measure (including some authors) [2] created and pilot tested a 96-item measure of community engagement in research. The new

measure included some items from the original, was fully quantitative and focused on 11 engagement principles. The 96-item measure assessed items on two scales—48 questions measuring quality (how well) and 48 questions measuring quantity (how often), as measured by three to five quality and three to five quantity items that correspond with each engagement principle [2]. The measure used 5-point Likert scale response options. Further details on the original measure are published elsewhere [2].

Content validation and item reduction of the quantitative measure of stakeholder engagement was completed in 2017–2018 [18], using a Delphi Process. A Delphi Process involves administration of multiple rounds of individual online and/or in-person surveys, with participant feedback on aggregated group responses for each round until reaching majority agreement on issues [18]. A five-round, modified Delphi Process was used to reach consensus on engagement principles and items for inclusion, elimination, and revision [19, 20]. The number of survey items on each scale (quantity and quality) was reduced from 48 to 32. There were three to five quality and three to five quantity items corresponding with each engagement principle, each assessed for quality (how well the engagement activity/strategy was implemented or completed) and quantity (how often the engagement activity/strategy was implemented) (Tables 1 and 2). The items that emerged after content validation (Delphi process) were subjected to cognitive response interview.

## Procedures

The institutional review boards at Washington University in St. Louis and at New York University approved this study and the consent procedures used. Interviewers (n = 4) attended a training to ensure consistent interview and data collection procedures. Interviewers were female, led by a PhD psychologist (VLST), MPH-trained project manager (NA), and two MPH student research assistants (including NL). The training provided in-depth instruction in cognitive interviewing, as well as an orientation to the interview guide and protocol. The interviewers received instruction on the use of tablets to administer the cognitive response interview to assure consistency and ease of administration. Although tablets were used during the interview to capture survey item and quantitative question responses, computer-assisted personal interview software was not used, and participant qualitative responses were captured using a digital recorder.

Sixteen eligible participants completed the in-person, one-on-one interviews in study interview rooms. Verbal consent for participation was obtained from all participants after an information sheet was provided. In order to assure that respondents understood their role in cognitive interviews, we explained that the purpose of the interview was to identify problems with item wording and to help us modify the items to improve their use in community-engaged research. We emphasized their role in helping clarify the questions before administering the final survey to 500 participants.

The first author (VLST) and three research assistants (including NA, NL) conducted interviews. Interviewees were greeted by a project staff member, directed to the interview room, and introduced to the interviewer, if different. At the beginning of the session, the interviewer introduced herself, explained the study, and briefly described the use of the tablet, explained the study procedures, and answered any of the participant's questions. The interviewer read each question aloud and highlighted the availability of a paper version of the survey to ease the participant's review and consideration. After the participant answered an item, the interviewer completed the verbal probing.

We first administered the draft of the survey items in a standard fashion, followed by scripted open-ended probes and spontaneous probes as needed for clarification. The

**Table 1. Summary of items removed throughout measure development & validation process–Delphi rounds 1 & 2.**

| Orig. EP[1] | New EP[2] | Delphi Round 1 | | Delphi Round 2 | |
|---|---|---|---|---|---|
| | | # of Items | Items | # of Items | Items |
| 1 | 1 | 0 | - | 0 | - |
| 2 | - | 4 | 1.) Show appreciation for community time and effort | - | - |
| | | | 2.) Highlight the community's involvement | | |
| | | | 3.) Give credit to community members and others for work | | |
| | | | 4.) Value community perspectives | | |
| 3 | - | 3 | 1.) Help community members with problems of their own | - | - |
| | | | 2.) Get findings and information to community members | | |
| | | | 3.) Help community members disseminate information using community publications | | |
| 4 | 2 | 1 | 1.) Ask community members for input | 0 | - |
| 5 | 3 | 0 | - | 2 | 1.) Seek input from all partners at every stage of the process |
| | | | | | 2.) Seek help from all partners at every stage of the process |
| 6 | 4 | 1 | 1.) Help community members achieve social, educational, or economic goals | 1 | 1.) Help all partners gain important skills from involvement |
| 7 | 5 | 0 | - | 1 | *Combined 2 Items into 1 Item*: |
| | | | | | 1.) Build on strengths within the community/ target population |
| | | | | | 2.) Build on resources within the community/ target population |
| 8 | 6 | 2 | 1.) Demonstrate that community members are really needed to do a good job | 2 | 1.) Demonstrate how all partners' ideas improve the work |
| | | | 2.) Enable community members to voice disagreements | | 2.) Make final decisions that reflect the ideas of all partners involved |
| 9 | - | 1 | 1.) Demonstrate that community members' ideas are just as important as academics' ideas | - | - |
| 10 | 7 | 1 | 1.) Include community members in plans for sharing findings | 1 | 1.) Share the results of how things turned out with all partners |
| 11 | - | 3 | 1.) Make plans for community-engaged activities to continue for many years | - | - |
| | | | 2.) Make commitments in communities that are long-term | | |
| | | | 3.) Want to work with community members for many years | | |
| - | 8 | - | - | 0 | - |
| *Total* | | 16 | - | 7 | - |

[1] EP1: Focus on local relevance and social determinants of health; EP2: Acknowledge the community; EP3: Disseminate findings and knowledge gained to all partners; EP4: Seek and use the input of community partners; EP5: Involve a cyclical and iterative process in pursuit of objectives; EP6: Foster co-learning, capacity building, and co-benefit for all partners; EP7: Build on strengths and resources within the community; EP8: Facilitate collaborative, equitable partnerships; EP9: Integrate and achieve a balance of all partners; EP10: Involve all partners in the dissemination process; EP11: Plan for a long-term process and commitment.

[2] EP1: Focus on community perspectives and determinants of health; EP2: Partner input is vital; EP3: Partnership sustainability to meet goals and objectives; EP4: Foster co-learning, capacity building, and co-benefit for all partners; EP5: Build on strengths and resources within the community or patient population; EP6: Facilitate collaborative, equitable partnerships; EP7: Involve all partners in the dissemination process; EP8: Build and maintain trust in the partnership.

interview probes were systematically developed before the interview, in order to search for potential problems (i.e., proactive verbal probes) [9] with survey items, as well as response options. The interview probes included questions and statements addressing the following:

**Table 2. Summary of items removed throughout measure development & validation process–Delphi rounds 3–4.**

| EP[1] | Delphi Round 3 | | Delphi Round 4 | |
|---|---|---|---|---|
| | # of Items | Items | # of Items | Items |
| 1 | 0 | - | 0 | - |
| 2 | 0 | - | 1 | 1.) Create a shared decision making structure |
| 3 | 0 | - | 0 | - |
| 4 | 0 | - | 0 | - |
| 5 | 1 | 1.) Help to fill gaps in community/patient population's strengths and resources | 0 | - |
| 6 | 1 | 1.) Foster collaborations in which all partners have input | 1 | 1.) Enable all people involved to voice their views |
| 7 | 2 | 1.) Interested partners are involved with sharing findings | 0 | - |
| | | 2.) The partners meet to communicate about the project | | |
| 8 | 0 | - | 0 | - |
| *Total* | 4 | - | 2 | - |

[1] EP1: Focus on community perspectives and determinants of health; EP2: Partner input is vital; EP3: Partnership sustainability to meet goals and objectives; EP4: Foster co-learning, capacity building, and co-benefit for all partners; EP5: Build on strengths and resources within the community or patient population; EP6: Facilitate collaborative, equitable partnerships; EP7: Involve all partners in the dissemination process; EP8: Build and maintain trust in the partnership.

- What do you think the statement is discussing/ describing?

- How would you rephrase the statement in your own words?

- Identify all the words in the statement, if any, that you do not understand.

- Rate how difficult it was to choose a response option for this statement.

- What, if anything, made this item difficult to answer?

- Rate the importance of this item for measuring community engagement.

- Describe what, if anything, makes it important to measuring community engagement in your own words.

- Rate how satisfied you are with the response options and how would you revise the response options?

To minimize the impact that the order of questions had on the overall results, we used 4 different versions of the questionnaire, with each set containing 16 items from each scale. Participants were assigned an identification number and interview version by NA before the interview was conducted. All 90- to 120-minute interviews were digitally recorded, and each session's recording was professionally transcribed. Each individual received a $50 gift card for participation.

Data coding and analysis were completed based on both the digitally recorded and professionally transcribed interviews and the field notes from interviewers compiled by NA. Transcripts were reviewed by the lead author (VLST) and project manager (NA) but were not returned to participants for review. After reviewing the project goals, the content of the interviews, and the existing literature, the first author (VLST) developed a defined coding guide that prescribed rules and categories for identifying and recording content.

Because the quality (how well) and quantity (how often) items repeat, these questions were only asked about the quality items. The remaining interview focused on participant reaction to changes in the response options as the survey moved from assessing the quality of community-engaged research activities to assessing the frequency of community-engaged research activities (quantity). For example, the interviewers asked:

- How did the change of scale, from a quality to quantity scale, affect your understanding of the item?

- Did it make the statement more difficult or easier to understand?

  Two queries were repeated to specifically address the quantity response option.

- "Tell me about your thought process when choosing a response option."

- "Tell me what you thought about to come up with your responses to the statements."

During the analysis phase, codes and themes were developed based on the elements deemed important in the cognitive response literature on item and questionnaire development [7–10]. The segments of text containing codes were identified and the codes were extracted, categorized and classified. All transcripts were coded. Once saturation was achieved by two coders, the senior investigator (VLST) and a research assistant (NL) read and coded the interview transcripts independently, identifying text units that addressed item clarity, literacy concerns, contextual issues, difficulty of the item, difficulty of item response, relevance and necessity for measurement of stakeholder engagement, in addition to clarity and appropriateness of response options. Coders met to reach a consensus on the definitions and examples used to code interview text from each transcript as the process proceeded. In cases of disagreement, the coders discussed discrepancies to reach consensus. On completion of coding, the coders reconvened to formulate core ideas and general themes that emerged from each interview. An interview summary, with examples, was developed for each theme. The full research team reviewed and discussed the themes identified in an effort to develop connections among themes and to clarify the relevance and importance of the findings for the measure and the field. Participants did not provide feedback on the findings.

## Results

The majority of the cognitive interview participants were female (n = 13; 81%), were African American (n = 11; 69%), and had a college degree or higher level of education (n = 9; 56%). Participants ranged in age from 24 to 73 years with a mean age of 47.3 years (Table 3). All of the participants had previous experience with community-engaged research.

**Table 3. Demographic characteristics of interview participants.**

| Demographic Characteristic N | | | | |
|---|---|---|---|---|
| **MEAN AGE** (N = 16): 47.3 (Range: 24 to 73) | | | | |
| **EDUCATION** (N = 16) | | | | |
| High School | 2 | | | |
| Some college or associate degree | 4 | | | |
| College degree | 2 | | | |
| Graduate Degree | 7 | | | |
| Missing | 1 | | | |
| **RACE** | **Male** | **Female** | **Missing** | **Total** |
| African American/ Black | 1 | 10 | 0 | 11 |
| White | 1 | 3 | 0 | 4 |
| Missing | 0 | 0 | 1 | 1 |
| Total | 2 | 13 | 1 | 16 |

## Item comprehension

Most participants did not readily report difficulties with the comprehension or definition of words or phrases; however, there were a few exceptions. Although not a concern among most participants, several reported that the wording of items addressing publication of research products was difficult, including wording related to dissemination, dissemination activities, and intellectual property. One participant stated, "Okay, now you put a big word in there. Okay, involve interested partners in dissemination activities." Terminology that addressed procedures to assure adherence to CBPR principles, such as *memorandum of understanding*, *governance*, *management responsibility*, and *mutually agreed upon* were among the terms identified as literacy concerns as stated by a participant: "I've not heard that one. Memorandum of understanding." *Stakeholder* was identified as jargon that presented a literacy issue. Participants' recommendations resulted in the replacement of the term stakeholder with *partners*. Several other words were identified that may also relate to the use of disciplinary jargon. For example, participants noted that food access could be simplified to "places to get food." Other words were unfamiliar and presented problems, such as *fosters*, *equitable*, and *inclusiveness* (See Table 4). Participants suggested plain language alternatives, such as *sharing results* or *sharing data* versus dissemination, *roles and responsibilities* versus memorandum of understanding and *articles and presentations* versus intellectual property.

Definitional issues also emerged. The terms *cultural factors*, *problem solving*, and *leadership responsibilities* were viewed as too general or ambiguous.

> "That can mean a lot of different things to a lot of different people. You can call a lot of different things a culture. I guess, to make it more clear, explaining what they mean by cultural." (Participant 1: Female, 54)

> "I don't know what you mean by problem solving. I don't know what you mean by ongoing. I don't know if there's a time limit on that or boundaries." (Participant 15: Female, 62)

Even when words such as *resources*, *environment*, and *partners* were understood, community participants wanted specifics and context, including what resources and which partners. Participants recommended the use of *all partners* to assure that both academic and community partners were considered, while sometimes noting that alternative wording was sometimes difficult to generate, particularly without context.

> "I would say I'm not completely clear what they mean by the environment." (Participant 1: Female, 54)

## Item response

Participants were generally satisfied with the response options (81.25% for quality and 87.5% quantity) and used the full range of response options for both scales (Table 5). Most participants indicated that it was "extremely easy" or "somewhat easy" (average 74.2% per item) to respond to the items tested. Participants noted that it was easy to transition between quality and quantity items and easier to respond to the quantity compared to the quality items.

Some participants requested the inclusion of "unsure, undecided" as a response option and the inclusion of numbers to ground the quantity scale. The larger concerns raised by participants related to the following: 1) the question stem that preceded items and 2) the ability to respond to items viewed as complex. The stem, "Please rate the quality/quantity or how well/how often academic partners do each of the following," created confusion about who

**Table 4. Summary of cognitive interview analyses: factors related to comprehension, response and suggestions for change.**

| Meaning | | |
|---|---|---|
| Literacy: self & others | | Comments/Suggestions |
| | Dissemination, dissemination activities, disseminate | Sharing results or sharing data |
| | Memorandum of Understanding | Roles and responsibilities |
| | Intellectual property | Articles and presentations |
| | Management responsibility | |
| | Accountable | |
| | Inclusion, inclusiveness, inclusive quality | |
| | Representation | |
| | Collaboration, collaborative | |
| | Equitable | |
| | Coalition | |
| | Fosters | Encourages, supports |
| | Incorporate factors | Delete and use examples |
| | Capacity | |
| | Governance | |
| | Mutually agreed upon, agreed-upon | |
| | Food access | Places to buy or get food |
| | Stakeholder (termed jargon) | |
| Vague | | |
| | Culture, cultural factors | (examples, context) |
| | Issues | (examples, context) |
| | Plan | (context) |
| | Problem Solving | |
| | Resources | (what resources, context) |
| | Capacity | (context) |
| | Environment | (context) |
| | Partner, partners, academic partners (there seems to be a desire to specify "all" | All partners, less confusion; who is included |
| | Leadership responsibility (also listed as preferred) | |
| **RESPONSES** | | |
| Quality Stem | Confusing, only specifies academic researchers; too wordy | Take word *quality* out of stem; specify all partners. |
| Recommended Changes | Additions | "Unsure, undecided," numbers to ground the quantity scale |
| Complex Questions | All partners assist in establishing roles and responsibilities for the collaboration | Reference single issue |
| | All partners have the opportunity to share ideas, input, leadership responsibilities, and governance (for example—memorandum of understanding, bylaws, organizational structure) as appropriate for the project. | It is always appropriate to share information in the partnership; too wordy |
| | Incorporate factors (for example—housing, transportation, food access, education, employment) that influence health status, as appropriate. | Delete "as appropriate." Always appropriate. |
| | Examine data together to determine the health problems that most people in the community think are important. | All partners look at the data to determine the health problems the community thinks are important. |
| | Partners agree on ownership and management responsibility of data and intellectual property. | Partners agree on ownership of data for publications and presentations. |
| **Importance** | | |
| Not important | Community has confidence they will receive credit for their contributions. | Researcher focused |
| | All partners have the opportunity to be coauthors when the work is published. | |
| Factors cited for importance | Trust, benefit, respect, power, control, decision making (mutual), value community | |

**Table 5. Average of participants (N = 16) choosing response options over all items.**

| Quantity Response | | Quality Responses | | Difficulty of Choosing Response* | |
|---|---|---|---|---|---|
| Response Option | Average, % | Response Option | Average, % | Response Option | Average, % |
| Never | 5.1 | Poor | 9.4 | Extremely Easy | 48.1 |
| Rarely | 9.0 | Fair | 8.6 | Somewhat Easy | 26.1 |
| Sometimes | 27.0 | Good | 19.2 | Neither Easy nor Difficult | 9.8 |
| Often | 36.5 | Very Good | 26.8 | Somewhat Difficult | 14.9 |
| Always | 22.4 | Excellent | 35.9 | Extremely Difficult | 1.2 |

*Asked after responding to item using quality scale.

subsequent items referenced when terms such as partners, partnership, or stakeholders were used. The stem is, therefore, implicated in the comprehension concerns noted. Participants recommended that the terms *quality* and *quantity* be removed from the stem and that the term *all partners* be used instead of the term *academic partners*.

Because of compound or "double-barreled questions," two items were identified as creating difficulty in responding: "All partners assist in establishing roles and responsibilities for the collaboration" and "Partners agree on ownership and management responsibility of data and intellectual property." These items were changed to "All partners assist in establishing roles and related responsibilities for the partnership" and "All partners agree on ownership of data for publications and presentations," respectively. Other items were identified as problematic because of strong beliefs about how health research should be conducted and partnerships managed.

## Relevance for community engagement

Two items were identified as unimportant to the assessment of community engagement by some participants. These items focused on items associated with the CBPR principle that addresses dissemination; thus, they were characterized as research focused. A participant's thoughts are illustrated below:

"I don't think it's a significant indicator of how engaged investigators are if they give authorship to the community partners. I think as long as they give credit to the community partners, that's what is important." (Participant 12: Male, 43)

The items seen as having the greatest relevance for the assessment of community engagement were trust, community benefit, respect, power/control, mutual decision making, and valuing the community. Sample participant explanations of principles and items appear below:

"Well, potentially the most important thing is identifying the issues that matter because if the issue itself doesn't matter then why would the community want to be engaged in the research if that's not important to them. Also, the result is going to be unimportant." (Participant 2: Female, 31)

"—that should be equal, the responsibilities. I'm thinking in terms of the community—well, as an equal relationship, so it's important that both are empowered to do what is necessary to better their circumstances." (Participant 5: Male, 73)

"So, yes, basically everything that both sides bring are being considered important because it gives mutual respect." (Participant 11: Female, 38)

"I think having trust among community levels—or amongst community members is important, because then you're going to get the most accurate answers and you're going to get—you're going to get even more than what you asked for. (Participant 7: Female, 24)

## Discussion

In order to understand the role that stakeholder-engaged research plays in the development, implementation, and outcomes of research studies, development and validation of measurement tools that can reliably and validly assess stakeholder engagement are required. This paper presents the results of one component in the measurement development process that also has implications for the way that we communicate with community partners about community-engaged research and the assessment of this work.

Results of cognitive response interviewing were consistent with concerns raised by Willis and Artino [9], who suggest that abstract terms are most problematic for participants. In this study, several terms commonly encountered in community engagement literature and measures were perceived as barriers and affected how community members responded to the item. Academic partners and researchers should likely guard against the assumption of common understanding, as participants considered some terms to be vague and in need of examples or context. Although it is appropriate to discuss culture, problem solving, plans, and environment, we must clarify what is referenced at specific times and with specific stakeholders. Even academics involved in community-engaged research may fail to realize when a common vocabulary has ceased to exist. In addition, plain language should be used to assure comprehension of discussions of publication and shared findings, the role of social determinants (such as the ability to get food), and efforts to assure that all partners are treated fairly and included in decisions and access to resources.

The findings suggest that item construction and comprehension issues were of greater concern in this measure development effort than response options. Most participants were satisfied with response options, found it easy to respond, and used the range of response options. Items that were excessively wordy and appeared to ask questions that required a response to two issues were identified as obstacles to participant response. It is important to note that the effort to develop consensus on items during the Delphi process described previously resulted in the development of some of the items identified as complex. Efforts to address diverse community input during item development may result in the need for additional review and editing to avoid item construction errors. In addition, the findings suggest that strong opinions and attitudes about an issue generated some concerns about the language used in survey items. This does not mean that an item should be reworded, but it does suggest that communication in partnerships should consider how messaging may affect dialogue and responses.

Few tested items were perceived as inappropriate or unimportant to the assessment of community-engaged research, although some participants questioned the importance of dissemination issues for community members versus academics. It is possible that the engagement principle guiding dissemination and the relevant items are sensitive to the research phase, i.e. more relevant to participants who are engaged in projects that are in or near the dissemination of the collaborative effort. The general acceptability of items suggests that the principles used to guide item selection are acceptable to the community members likely to be encountered or to participate in stakeholder-engaged research and assessment [19], although participants suggested changes in words and terms, as well as item structure. Minimal concerns were related to response options. These findings should be interpreted cautiously because of the small sample size. However, cognitive response interviewing [7,10] provides in-depth insight into how participants are thinking about and interpreting items, the factors that affect their

interpretation and responses, and how comfortable they feel with the language, options, and coverage of topics important to an issue.

## Conclusions

Understanding how the level of engagement in a partnership is developing and to what extent level of engagement is a predictor of outcomes in stakeholder-engaged research is important to making progress in community-engaged research. Because researchers have suggested that research on measures of stakeholder engagement is not very strong [3] and rigorous measurement of engagement is required, the results of the current study contribute to an effort to develop and validate a broadly applicable measure of stakeholder engagement. In the results of the cognitive response interviews, which were used to refine the questionnaire being developed, participants suggested concerns about plain language, literacy and clarity of question focus, and the lack of context clues to facilitate responses to items that query research experience. Given that the presented findings are consistent with the literature on stakeholder engagement [2, 16]—although communication concerns were highlighted in the current study —these findings should be of use to both those assessing community-engaged research and those engaging the community in the research process. Researchers should remain cognizant of the use of plain language, literacy levels, and contextual cues as partnerships are discussed and as agreements are developed.

## Acknowledgments

The authors thank Sharese Willis for her help editing the manuscript.

## Author Contributions

**Conceptualization:** Vetta L. Sanders Thompson, Deborah J. Bowen, Melody S. Goodman.

**Data curation:** Vetta L. Sanders Thompson, Nora Leahy, Nicole Ackermann.

**Formal analysis:** Vetta L. Sanders Thompson, Nora Leahy.

**Methodology:** Vetta L. Sanders Thompson, Melody S. Goodman.

**Project administration:** Nicole Ackermann.

**Resources:** Melody S. Goodman.

**Writing – original draft:** Vetta L. Sanders Thompson, Nicole Ackermann.

**Writing – review & editing:** Vetta L. Sanders Thompson, Nora Leahy, Nicole Ackermann, Deborah J. Bowen, Melody S. Goodman.

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
