## [Decision Letter · Decision Letter 0]

2 Sep 2020

PONE-D-20-04537

Community partners’ responses to items assessing stakeholder engagement:  Cognitive response testing in measure development

PLOS ONE

Dear Dr. Thompson,

Thank you for submitting your manuscript to PLOS ONE. After careful consideration, we feel that it has merit but does not fully meet PLOS ONE’s publication criteria as it currently stands. Therefore, we invite you to submit a revised version of the manuscript that addresses the points raised during the review process.

Please note that reviewer number 2 and 3 are the same person. The request was sent twice, and he or she answered likewise. I have kept the two recommendations because there are few different comments at the end of both reviews that might be useful for you.

I recommend to take a careful consideration of the thoughtful revision made by both reviewers, specially of reviewer number 1. They are asking for minor changes, but they might clarify the text and make it more accessible for the readership.

We look forward to receiving your revised manuscript.

Kind regards,

Roxanna Morote Rios, Ph.D

Academic Editor

PLOS ONE

Journal Requirements:

2. Please provide additional details regarding participant consent. In the ethics statement in the Methods and online submission information, please ensure that you have specified how verbal consent was documented and witnessed.

3. Please refer to any sample size calculations performed prior to participant recruitment. If these were not performed please justify the reasons or cite similar literature. Please refer to our statistical reporting guidelines for assistance (https://journals.plos.org/plosone/s/submission-guidelines.#loc-statistical-reporting).

4.We note that you have indicated that data from this study are available upon request. PLOS only allows data to be available upon request if there are legal or ethical restrictions on sharing data publicly. For information on unacceptable data access restrictions, please see http://journals.plos.org/plosone/s/data-availability#loc-unacceptable-data-access-restrictions.

Reviewers' comments:

Reviewer's Responses to Questions

**Comments to the Author**

1. Is the manuscript technically sound, and do the data support the conclusions?

Reviewer #1: Yes

Reviewer #2: Yes

Reviewer #3: Yes

2. Has the statistical analysis been performed appropriately and rigorously? 

Reviewer #1: Yes

Reviewer #2: Yes

Reviewer #3: Yes

3. Have the authors made all data underlying the findings in their manuscript fully available?

Reviewer #1: Yes

Reviewer #2: Yes

Reviewer #3: Yes

4. Is the manuscript presented in an intelligible fashion and written in standard English?

Reviewer #1: Yes

Reviewer #2: Yes

Reviewer #3: Yes

5. Review Comments to the Author

Reviewer #1: Overall the paper makes a meaningful contribution, especially considering the need for qualitative evaluations of quantitative measures. The experience of the survey-taker, participant, or stakeholder (whatever the nomenclature) should be an important part of the review process. However, it is not a completed product. There are some issues in the paper that should be resolved before moving forward with the manuscript. For example, there are several relevant areas lacking clarity, especially in the consistent use of terminology, and this lack of clarity undercuts the goals of the paper.

The clarity issue is exemplified by the discussion under the Item Selection section of the paper (ll. 124-147). An original survey is discussed that has 60 items, then in the next paragraph the authors test a 96-item version. Are these two versions of the same survey? If not, please state in a straightforward fashion that they are two different surveys. If so, how and why did the survey grow to 96 items? The last paragraph of that section states that the number of survey items was narrowed from 48 to 32. Which 48? Is this the 48 quantitative or the 48 qualitative items? Or is it another subset of the 96 items? Or are these 48 items unrelated to the 96 items mentioned earlier? Also in this discussion, a Delphi process is introduced. The process either should be defined briefly right after first mention or covered in an appendix or footnote. Given the issues with the number of items and how items were chosen, the discussion of "16 items from each scale total" on line 180 is likely to cause further confusion.

As noted, the paper has potential to make a contribution to the literature on measurement, cognitive interviewing, and community engagement. The paper is at its strongest when the authors describe that contribution in the second paragraph of the paper (ll. 65-75). This description is clear and to the point; it should serve as a model for how to convey your other important points in the paper. The other areas that require revision for clarity, typos, or other reasons are listed below by page and line number.

-Page 3, line 57: Citations demonstrating the increased interest in community-engaged research would strengthen your opening argument.

-I recommend moving the first sentence on line 87 to the previous paragraph before "One approach to identifying..." Then the following paragraph would begin with "Cognitive interview methods..." The point of the "Researchers recognize..." sentence sets up the last sentence of the preceding paragraph better than it does the following paragraph.

-Page 5, line 106: missing quotation marks at the end of the sentence.

-Page 6: item number issues already noted above

-Page 8, line 184: This would be a good place to insert the actual probes, which are discussed in the preceding and following paragraphs. Even if only a subset are listed or described, this would help the reader have a better understanding of the nature of the interviews.

-Page 9, line 190-191: The sentence starting with "Behavioral coding..." is confusing. Are the following items (ll. 192-201) the probes (see point above)? Coding is the researchers' process, not an interaction with a participant. But this sentence implies that behavioral coding involved the participant.

-Page 10, line 211: The first sentence discusses relevant codes. What is the threshold for relevance? How did you arrive at that? Did you follow best practices, prior research, etc.? It is important to be clear what overall guiding principle was employed because the codes and themes covered here are the keystone of the whole paper.

-Page 10, lines 212-222: Very good section. This description of the process was clear and helpful.

-Page 10, line 225: Extra space not needed before Results section.

-Table 2, page 12: Helpful, informative table.

-Page 14, line 274: Can get rid of this quote. It's already covered in the preceding sentences.

-Page 17, line 332: The authors say that the items were difficult to comprehend. I would be careful here. You didn't demonstrate that (or maybe you did but it isn't demonstrated in the paper); what you showed was that the wording was a hurdle that affected how community members responded to the item. Those are two different things.

Reviewer #2: The primary objective of this study is clearly defined and highly relevant to studies requiring stakeholder engagement. The authors identify that measures of stakeholder engagement are not very strong methodologically. They add to the literature by addressing measurement of stakeholder engagement in terms of literacy concerns, attitudes about information needed to judge engagement, and response preferences for items used in public health community-engaged research. The methods clearly describe their approach which draws on 16 individuals for one on one cognitive response interviews. They clearly describe methods for refinement in Items from an initial survey containing 60 items to 48 and then to 32, using a modified Delphi process. Could a figure be added to the manuscript to show the research and decision path from the 60 items down to the 32 items.

The cognitive interview participants were diverse in age, education, and predominantly African American women. The results described the item response and the steps taken to remove items and to clarify questions that had raised issues for participants. Importantly, the results support the recommendation that academic partners and researchers should guard against the assumption of common understanding is participants found some items vague and needing more context. Comprehension issues were of greater concern in measure development than response options.

Accordingly, the details described in this paper demonstrate the rigor and refinement the authors bring to the issue of measuring community engagement. Their conclusions are well justified and should help advance the field.

The tables add to the manuscript.

The title for Table 2 might be expanded to give more context.

Reviewer #3: The primary objective of this study is clearly defined and highly relevant to studies requiring stakeholder engagement. The authors identify that measures of stakeholder engagement are not very strong methodologically. They add to the literature by addressing measurement of stakeholder engagement in terms of literacy concerns, attitudes about information needed to judge engagement, and response preferences for items used in public health community-engaged research. The methods clearly describe their approach which draws on 16 individuals for one on one cognitive response interviews. They clearly describe methods for refinement in Items from an initial survey containing 60 items to 48 and then to 32, using a modified Delphi process. Could a figure be added to the manuscript to show the research and decision path from the 60 items down to the 32 items.

The cognitive interview participants were diverse in age, education, and predominantly African American women. The results described the item response and the steps taken to remove items and to clarify questions that had raised issues for participants. Importantly, the results support the recommendation that academic partners and researchers should guard against the assumption of common understanding is participants found some items vague and needing more context. Comprehension issues were of greater concern in measure development than response options.

Accordingly, the details described in this paper demonstrate the rigor and refinement the authors bring to the issue of measuring community engagement. Their conclusions are well justified and should help advance the field.

The tables add to the manuscript.

The title for Table 2 might be expanded to give more context.

6. PLOS authors have the option to publish the peer review history of their article (what does this mean?). If published, this will include your full peer review and any attached files.

Reviewer #1: No

Reviewer #2: No

Reviewer #3: No

---

## [Author Response · Author response to Decision Letter 0]

23 Sep 2020

Dear Reviewers:

Thank you for your review of this manuscript and your insightful questions, comments and suggestions. We have addressed the concerns and issues discussed and believe that the reviewer’s comments and suggestions strengthen the manuscript. We have outlined our responses below. We used track changes and all changes are highlighted in yellow.

Reviewer 1

1. We have addressed the clarity issue noted in the discussion under the Item Selection section of the paper (ll. 124-147). We have clarified that the original survey was 60 items and that a new survey was created in 2013, using some of these items and focused on coverage of CBPR principles, all of which are quantitative. This survey was 96 items. 

2. We go on to explain the effort to reduce the 96-item set in greater detail to provide clarity. We discuss the process of moving from 48- items to 32-items. 

3. As noted, the Delphi process was introduced without a definition. We have added a definition and the appropriate citation. 

4. Given the issues with the number of items and how items were chosen, we have edited the section for clarity to assure that the "16 items from each scale total" on line 180 is easier to understand. 

5. Page 3, line 57: Citations demonstrating the increased interest in community-engaged research have been added. P.3, line 57

6. We have changed the sentence positions to improve the set-up of the paragraphs as suggested p.4, line 85-89 

7. The missing quotation marks have been added on -Page 5, line 107. 

8. The item number issues discussed above have been edited on page 6 as well

9. We have deleted the probes from page 9 and inserted them on what is now Page 10, lines 196-208. 

10. We have rewritten the section on Page 11, lines 220-247 to delete references to behavioral coding...". We have also included the probes used to understand participant reactions to questions about response options. 

11. Page 11, line 244-247 we have deleted the reference to relevant codes and have instead explained how we went about developing the coding strategy, including prior research and deemed important to explore in the cognitive response literature. 

12. We have deleted the extra space before the Results section. Page 13, line 262

13. We have deleted the superfluous quote as suggested. -Page 17, line 309 

14. Your point is well taken and it is likely that we have overstated our findings related to participants’ abilities to comprehend particular words. We have reworded this sentence. Page 20, line 367-368. 

Reviewer 2

1. In addition to the text that clarifies the process of reducing the number of items in each scale, we have included a table (now Table 1) to show the research and decision path down to the 32 items. This has resulted in a renumbering of all tables. P. 7, 13, 15, 18

2. We have expanded the title for Table 2 to give more context. P. 13

We appreciate your feedback and the opportunity to improve this manuscript.

---

## [Editor Report · Decision Letter 1]

22 Oct 2020

Community partners’ responses to items assessing stakeholder engagement:  Cognitive response testing in measure development

PONE-D-20-04537R1

Dear Dr. Sanders Thompson

We’re pleased to inform you that your manuscript has been judged scientifically suitable for publication and will be formally accepted for publication once it meets all outstanding technical requirements.

Kind regards,

Roxanna Morote Rios, Ph.D

Academic Editor

PLOS ONE

---

## [Editor Report · Acceptance letter]

13 Nov 2020

PONE-D-20-04537R1 

Community Partners’ Responses to Items Assessing Stakeholder Engagement:  Cognitive Response Testing in Measure Development 

Dear Dr. Thompson:

I'm pleased to inform you that your manuscript has been deemed suitable for publication in PLOS ONE. Congratulations! Your manuscript is now with our production department. 

Kind regards, 

on behalf of

Dr. Roxanna Morote Rios 

Academic Editor

PLOS ONE